# Examining Specific Theory-of-Mind Aspects in Amnestic and Non-Amnestic Mild Cognitive Impairment: Their Relationships with Sleep Duration and Cognitive Planning

**DOI:** 10.3390/brainsci15010057

**Published:** 2025-01-10

**Authors:** Areti Batzikosta, Despina Moraitou, Paschalis Steiropoulos, Georgia Papantoniou, Georgios A. Kougioumtzis, Ioanna-Giannoula Katsouri, Maria Sofologi, Magda Tsolaki

**Affiliations:** 1Laboratory of Psychology, Department of Cognition, Brain and Behavior, School of Psychology, Faculty of Philosophy, Aristotle University of Thessaloniki (AUTh), 54124 Thessaloniki, Greece; aretiba@hotmail.com; 2Laboratory of Neurodegenerative Diseases, Center of Interdisciplinary Research and Innovation (CIRI-AUTH), Balcan Center, Buildings A & B, 57001 Thessaloniki, Greece; gpapanto@uoi.gr (G.P.); tsolakim1@gmail.com (M.T.); 3Department of Respiratory Medicine, Medical School, Democritus University of Thrace, 68100 Alexandroupolis, Greece; steiropoulos@yahoo.com; 4Laboratory of Psychology, Department of Early Childhood Education, School of Education, University of Ioannina, 45110 Ioannina, Greece; m.sofologi@uoi.gr; 5Institute of Humanities and Social Sciences, University Research Centre of Ioannina (URCI), 45110 Ioannina, Greece; 6Department of Turkish Studies and Modern Asian Studies, Faculty of Economic and Political Sciences, National and Kapodistrian University of Athens, 15772 Athens, Greece; gkougioum@ppp.uoa.gr; 7Department of Psychology, School of Health Sciences, Neapolis University Pafos, Pafos 8042, Cyprus; 8Department of Occupational Therapy, Faculty of Health and Caring Sciences, University of West Attica, 12243 Athens, Greece; ykatsouri@uniwa.gr; 9Greek Association of Alzheimer’s Disease and Related Disorders (GAADRD), Petrou Sindika 13 Street, 54643 Thessaloniki, Greece

**Keywords:** executive functions, non-literal speech comprehension, sleep, social cognition, rules’ violation, third-order ToM

## Abstract

**Background/Objectives:** The study examined the relationships between specific Theory-of-Mind (ToM) dimensions, cognitive planning, and sleep duration in aging adults. **Methods:** The sample included 179 participants, comprising 46 cognitively healthy individuals, 75 diagnosed with amnestic Mild Cognitive Impairment (aMCI), and 58 with non-amnestic (naMCI). The mean age of the participants was 70.23 years (SD = 4.74), with a mean educational attainment of 12.35 years (SD = 3.22) and gender distribution of 53 men and 126 women. ToM assessment included tasks measuring the understanding and interpretation of non-literal speech, proverbs and metaphors, as well as an emotion-recognition test. For cognitive planning, a Tower Test was utilized. Sleep duration was measured using actigraphy. **Results:** We identified significant differences in various ToM tasks’ performance between the groups, particularly in non-literal speech tasks and third-order ToM stories. The HC group consistently outperformed both MCI groups in these tasks, with aMCI showing higher performance than naMCI. Mediation analysis applied to examine potential direct and indirect effects of sleep duration on ToM tasks indicated that total sleep time had significant indirect effects through cognitive planning—mainly as rule violation total score—on specific ToM aspects. Hence, besides the effects of MCI pathologies and especially of naMCI, sleep duration seems also to be associated with ToM performance in aging via specific executive functioning decrements. **Conclusions:** The findings underscore the social implications of ToM deficits due to MCI and/or sleep duration decrease, particularly in naMCI older adults, as they can seriously impair their social interactions. Targeted interventions could improve emotional understanding, communication, and overall quality of life.

## 1. Introduction

This article examines the relationships between specific cognitive abilities, specific sleep parameters, and specific aspects of Theory of Mind (ToM) in cognitively healthy older adults and individuals with Mild Cognitive Impairment (MCI). The study analyzes differences between amnestic MCI (aMCI) and non-amnestic MCI (naMCI) subtypes and investigates potential associations among the variables of interest across the groups.

MCI is considered an intermediate state between normal aging and dementia, characterized by impairments in cognitive functions without significant disruption to daily activities [1]. The two primary subtypes of MCI are aMCI, which primarily affects memory, and naMCI, which involves other cognitive domains such as language, visuospatial abilities, or executive functions [2].

MCI not only impacts cognitive domains but also has significant implications for daily life and social functioning. Deficits in Τheory of Μind (ToM), particularly in understanding and predicting others’ emotions and intentions, can hinder effective communication and social interactions. These deficits may result in challenges with interpreting nuanced emotional cues, maintaining relationships, and engaging in effective social behavior. Recent studies have highlighted the practical consequences of ToM impairments in MCI, emphasizing the need for further exploration of these relationships [3,4]

Social cognition encompasses the capacity to identify, observe and interpret socially relevant information. It involves a wide array of abilities, including social knowledge and ToM [5,6]. ΤoM is the ability of an individual to understand and attribute mental states, such as beliefs, desires, intentions, and emotions toward others, as well as to predict and explain their behavior based on these mental states. ToM is a fundamental component of social cognition and is crucial for effective social interaction and communication in healthy and pathological lifespan development and aging [7]. It is usually divided into affective ToM (understanding emotions) and cognitive ToM (understanding thoughts and beliefs) [8].

Research in social and cognitive neuroscience has identified several key components of ToM. A fundamental aspect, known as ‘first-order’ ToM, involves the ability to infer another person’s mental states, such as beliefs and thoughts. ‘Second to five-order’ ToM are more complex forms that enable individuals to understand others’ perspectives on mental states, involving the simultaneous consideration of two viewpoints [9]. More advanced ToM processes, like interpreting non-literal speech, proverbs, and metaphors, require even higher levels of metarepresentation [10].

Recent studies support that a complex network of brain regions supports ToM by processing social information and enabling the understanding of others’ mental states [11]. Key regions involved include the prefrontal cortex, critical for analyzing complex social cues and attributing cognitive states; the superior temporal sulcus, which detects movements and intentions of others; and the anterior cingulate cortex, involved in emotional processing and empathy [12]. Additionally, the amygdala plays a central role in regulating emotions and responding to emotionally charged social situations. These regions, interacting with the medial prefrontal cortex, create a network that regulates emotions and enhances understanding of social cues, facilitating a comprehensive understanding of others’ cognitive states [13].

In individuals with MCI, significant difficulties emerge in interpreting non-literal language, including metaphors, proverbs, and humor. This understanding requires complex socio-cognitive processes, such as abstract thinking, ToM, and the ability to contextualize information—all of which are frequently compromised in MCI. Specifically, interpreting metaphors and proverbs, which demand both abstract thinking and the application of previous knowledge, can become especially challenging. In MCI, access to and integration of this contextual information may be disrupted, leading to an impaired grasp of implied meanings. As a result, MCI patients may miss the intended nuances of expressions, affecting their social interactions and overall communication effectiveness [14].

Recent studies using ToM assessments to examine socio-cognitive functioning in individuals with MCI reveal that challenges in interpreting metaphorical language and proverbs are closely linked to deficits in executive functioning and memory, two domains that seem to be heavily impacted by MCI [15]. A comprehensive 2022 review further emphasizes that ToM impairments are prevalent across various MCI subtypes, with difficulties in comprehension becoming more pronounced as cognitive decline advances [14].

At the neurophysiological level, recent studies highlight that reduced functionality in regions such as the prefrontal cortex (PFC) and temporal lobes is closely linked to difficulties in processing non-literal language in MCI. The PFC, which plays a crucial role in abstract thinking and ToM, often exhibits alterations in MCI, hampering patients’ abilities to interpret complex or indirect meanings [14]. Additionally, the connectivity between the PFC and the temporal lobes, which are essential for storing and retrieving knowledge, tends to weaken, further impacting comprehension and interpretation of figurative expressions. Humor and/or irony, which demand quick comprehension and real-time association of information, are particularly affected by reduced connectivity between these brain regions [16]. This is also evident in sarcasm comprehension, which involves complex social inference. A recent cross-sectional work of Tsentidou et al. [17] highlighted deficits in understanding sarcasm among individuals with MCI and vascular risk factors, compared to healthy controls, which may signal early ToM impairments linked to cardiovascular health. Furthermore, the longitudinal findings by Tsentidou et al. [7] revealed that individuals with vascular risk factors and MCI showed a significant decline in understanding non-literal speech over time, particularly in complex ToM tasks like those requiring sarcasm understanding. This decline highlights that the progression of social–cognitive impairments is linked to both cardiovascular problems and neurodegeneration in older adults. Current literature confirms that such neurophysiological changes are early signs of increased neurodegeneration [18], with implications for predicting the progression of potential preclinical stages of MCI to AD dementia [19].

Emotion recognition gradually declines over time as a result of normal aging processes. Brain regions responsible for emotion recognition, such as PFC, undergo age-related changes that affect the ability to recognize and interpret emotional expressions, particularly more complex ones, such as fear or sadness [20]. In individuals with MCI, these difficulties are more pronounced and emerge earlier compared to the general population. The neurodegenerative changes characteristic of MCI, including reduced connectivity between the PFC and temporal lobes, exacerbate impairments in processing emotional information [21]. This results in challenges in understanding nuanced expressions such as irony or sarcasm. These distinct deficits in emotion recognition serve as significant markers for monitoring the progression of cognitive decline and the potential transition to AD dementia.

Based on the extant literature, it seems that ToM is associated with cognitive control abilities, including inhibition, cognitive flexibility, and planning [22]. These functions are crucial for interpreting non-literal language. Specifically, executive functions such as cognitive flexibility, which integrates inhibitory control with set-shifting, are central to understanding metaphorical language and humor. In MCI, degenerative changes frequently occur in the prefrontal cortex, leading to diminished capabilities in these areas and impacting overall comprehension and social interaction. Inhibition, which allows the brain to filter out irrelevant information, is crucial for avoiding surface interpretations of metaphors and focusing on abstract meaning. Similarly, cognitive flexibility enables patients to shift cognitive sets and adapt to the context of non-literal language, as well as understand irony and humor, which require rapid transitions from literal to figurative or ironic perspectives [23]. The planning process is equally important, as it aids patients in anticipating outcomes and forming a comprehensive understanding of non-literal messages. Recent studies suggest that the weakening of these functions leads to deficits in social and communicative abilities among MCI patients, directly impacting their Theory of Mind and cognitive skills essential for logical and emotional comprehension of figurative language [22].

Sleep plays a critical role in ToM, as it supports various processes, which are essential for understanding the perspectives and emotions of others. Disruptions in sleep, such as sleep apnea and insomnia, negatively impact ToM abilities due to interruptions like intermittent hypoxia and fragmented sleep. These disturbances could impair executive functions such as planning, all of which are crucial for processing non-literal language like metaphors and irony, sarcasm or humor [24].

Furthermore, sleep deprivation has been shown to reduce prefrontal cortex (PFC) activity, which is central to both ToM and executive functions. The PFC’s connectivity is weakened during sleep deprivation. This has been suggested to disrupt the ability to engage in perspective-taking, leading to difficulties in social interactions and the interpretation of social cues [25]. Studies suggest that adequate sleep, particularly during stages like REM and slow-wave sleep (SWS), helps integrate experiences relevant to social cognition and supports emotional regulation, which is essential for effective social communication and empathy [26].

As regards the potential effects of sleep disturbances on ToM in MCI, sleep has been supported to play a significant role [27]. In this case, cognitive vulnerabilities due both to MCI and sleep problems exacerbate the challenges in interpreting social cues and engaging in complex social interactions [28,29]. Different types of MCI seem to affect ToM in different ways [30]. Regarding amnestic MCI, studies indicate that patients tend to exhibit moderate difficulties in ToM skills, mainly due to memory problems. However, in the early stages, these issues may be mild. Individuals with non-amnestic MCI, especially when executive functions are impaired due to MCI-related pathology and potential sleep problems, may exhibit more pronounced deficits in ToM. Since ToM has been supported to require complex cognitive processing and often relies on executive functions such as planning, this subtype may lead to greater impairments in social skills [31].

Studies also suggest that sleep deprivation can affect emotion regulation, another key component of ToM [32]. Impaired emotion regulation makes it more difficult for individuals to interpret and respond appropriately to others’ emotional states, contributing to poor social functioning [33]. Recent research further supports the idea that improved sleep quality could help mitigate some of the cognitive and social difficulties associated with MCI by enhancing executive functions and ToM abilities directly [28,29,34]. This highlights the potentially complex relationship between sleep, cognitive control, and social cognition in older adults, particularly those at risk of AD dementia [30,35].

As regards the potential associations between sleep parameters, cognitive control abilities and specific ToM aspects, based on the findings from the first part of this study [36], individuals with aMCI and naMCI had only one difference with healthy controls in sleep parameters. They exhibited shorter sleep duration compared to healthy controls and between each other (aMCI > naMCI), consistent with prior studies suggesting a link between reduced sleep duration and cognitive decline. In fact, from the main cognitive control abilities examined in the study, only cognitive planning showed deficits in both MCI groups compared to healthy controls, with significant correlations between sleep duration and performances on the task requiring cognitive planning.

In light of these findings and the extant literature, the present part of our cross-sectional study aimed to examine the performance of aMCI and naMCI patients, as compared to healthy controls, in specific tasks requiring higher-order ToM abilities, which are less studied formerly, and explore the relationships—if any—between these specific abilities, cognitive planning, and sleep duration, to form a clear picture for ToM abilities that can be affected by amnestic and non-amnestic MCI, and can deteriorate due to a specific sleep disturbance.

### Aims and Hypotheses of the Study

The study aimed to examine potential associations between sleep duration, various complex ToM abilities, and cognitive planning in aMCI and naMCI subtypes.

The Following Hypotheses Were Formulated:

**Hypothesis** **1a.**
*Individuals with MCI (both aMCI and naMCI) would exhibit poorer ToM performance as regards various aspects of ToM compared to healthy controls.*


**Hypothesis** **1b.**
*Different subtypes of MCI would exhibit distinct patterns of impairment in various ToM skills. Specifically, we expected that individuals with naMCI would show more pronounced impairments, at least in specific ToM skills, than individuals with aMCI.*


**Hypothesis** **2a.**
*Beyond and above the potential effects of MCI pathologies, reduced sleep duration, which was found to be the only sleep measure that is differentiated between healthy older adults, aMCI and naMCI patients, would be associated with lower performance at least on some of the specific ToM tests [36].*


**Hypothesis** **2b.**
*Beyond and above the potential effects of MCI pathologies, the relationships between reduced sleep duration and performance in ToM tasks would be mediated by cognitive planning, which was found to be significantly correlated with sleep duration [36].*


## 2. Materials and Methods

### 2.1. Design

This study was the first phase (cross-sectional) of a broader longitudinal project, comparing three groups: (a) cognitively healthy older adults (HC ≥ 65 years), (b) people with aMCI, (c) people with naMCI, in specific sleep parameters, specific cognitive control abilities, specific aspects of ToM, and their relationships.

### 2.2. Participants

Power analysis was conducted using G*Power [37] to determine the appropriate sample size for the study, revealing that a minimum of 148 participants was necessary to achieve a power level of 0.80. Ultimately, 185 individuals were recruited and assessed for cognitive health. Among these, one participant was identified with Subjective Cognitive Impairment (SCI), another was diagnosed with early dementia, and four were receiving treatment for sleep disorders. Consequently, these six individuals were excluded from the analysis.

The final sample consisted of 179 participants, which included 46 cognitively healthy individuals, 75 diagnosed with aMCI, and 58 diagnosed with naMCI. The sample comprised 53 men and 126 women, with a mean age of 70.23 years (SD = 4.74) and a mean educational attainment of 12.35 years (SD = 3.22).

To be eligible for inclusion in the study, participants were required to be over 65 years of age and have completed at least six years of formal education. Participants were classified into categories of healthy cognitive status, aMCI, or naMCI following a comprehensive neuropsychological assessment, which adhered to Petersen’s diagnostic criteria [38] and the Diagnostic and Statistical Manual of Mental Disorders, Fifth Edition, Text Revision (DSM-5-TR) [39].

### 2.3. Inclusion Criteria

The control group comprised 46 healthy adult volunteers living in the community, all demonstrating excellent cognitive health as indicated by their Montreal Cognitive Assessment (MoCA) scores, which ranged from 27 to 30 (M = 27.8, SD = 0.74).

The aMCI group consisted of 75 adults who had been diagnosed with MCI within the past two years. Their diagnosis adhered to the DSM-5-TR criteria for Mild Neurocognitive Disorders [39]. This diagnosis was further corroborated through a comprehensive neuropsychological assessment, neurological examinations, neuroimaging studies, and psychiatric evaluations. The inclusion criteria for this group included (a) a diagnosis of Minor Neurocognitive Disorder and (b) a score of at least 1.5 standard deviations (SDs) below the normative mean on an episodic memory test. Following these evaluations, all 75 participants in the aMCI group met the criteria for the amnestic subtype, with their MoCA scores ranging from 24 to 29 (M = 24.59, SD = 2.7).

The naMCI group consisted of 58 adults diagnosed with MCI over the last two years. Like the aMCI group, the inclusion criteria for the naMCI participants included (a) a diagnosis of Minor Neurocognitive Disorder and (b) a score of at least 1.5 standard deviations (SDs) below the normal mean in at least one cognitive domain other than memory, based on the neuropsychological assessments. According to the evaluations, 58 participants were diagnosed with the non-amnestic subtype, with their MoCA scores ranging from 20 to 29 (M = 26.10, SD = 2.47).

### 2.4. Exclusion Criteria

The exclusion criteria included (a) a history of psychiatric disorder, (b) substance abuse or alcoholism, (c) prior traumatic brain injuries, (d) neurological disorders, (e) diagnosed sleep disorders, (f) use of medications for sleep disorders, (g) medications for depression or anxiety, and (h) the presence of cognitive complaints in the healthy control group.

A comprehensive neuropsychological evaluation was conducted for all participants to establish their diagnostic status at the Greek Association of Alzheimer’s Disease and Related Disorders. This assessment utilized several validated instruments. To screen for affective disorders, participants completed the Geriatric Depression Scale [40,41], the Beck Depression Inventory [42], the Beck Anxiety Inventory [43], and the Short Anxiety Screening [44,45]. The Neuropsychiatric Inventory [46,47] was used to identify any neuropsychiatric symptoms.

Cognitive functioning was assessed with the Mini-Mental State Examination (MMSE) [48,49] and the Montreal Cognitive Assessment (MoCA) [50,51] to evaluate overall cognitive status. Furthermore, the Functional Cognitive Assessment [52] assessed executive functions through tasks related to daily living activities. A range of standardized cognitive tests was administered to evaluate memory, attention, executive functions, and language skills. The Global Deterioration Scale (GDS) [53] was applied to gauge the progression of cognitive decline, where stage 1 represents no cognitive decline and normal functioning, and stage 3 indicates MCI. For a detailed overview of the neuropsychological tests utilized, please see Tsolaki et al. [54].

Statistical analyses indicated no significant differences among the three groups regarding age, F (2, 176) = 2.977, *p* = 0.054; years of education, F (2, 176) = 0.452, *p* = 0.637; and gender distribution, χ^2^ = 1.849, *p* = 0.397.

### 2.5. Ethics

Participants were informed about the aims of the study through both oral and written communication, and they were assured of the confidentiality of their data. They provided written consent to indicate their voluntary participation, and they had the right to withdraw from the study at any time. The collection of demographic information, including age, gender, and education, was conducted in accordance with European Union legislation (28 May 2018), which permits the use of sensitive personal data for research purposes. Participants were also informed that they could request the removal of their data from the web database in writing. The research protocol received approval from the Scientific and Ethics Committee of the Greek Association of Alzheimer’s Disease and Related Disorders (Approval Code: 29/15-02-2017) and adhered to the guidelines outlined in the Declaration of Helsinki.

### 2.6. Procedure

Undergraduate psychology interns facilitated participant recruitment from the Daycare Centers of the Greek Association of Alzheimer’s Disease and Related Disorders, as well as from the Aristotle University of Thessaloniki. Volunteers who met the inclusion criteria were invited to participate, and those who agreed were subsequently contacted by the study psychologist. The psychologist provided detailed information regarding the study’s purpose and procedures. Participants then scheduled two morning appointments within one week, each lasting up to one hour, to complete the assessments. Written informed consent was obtained during the first appointment, ensuring confidentiality and clarifying the study’s objectives. The tests were administered individually in a quiet, comfortable environment, utilizing two different versions of the testing battery to mitigate order effects. Participants did not receive any compensation for their involvement.

### 2.7. Instruments

#### 2.7.1. Theory of Mind

For the longitudinal phase of our broader empirical work, Natsopoulos’ Theory-of-Mind battery [55] was used, which evaluates various aspects of ToM, divided into three similar sub-batteries to minimize potential order and practice effects as much as possible. Scoring was conducted later by the examiner. Specifically, in the present study, which represents the cross-sectional part of the broader empirical word, 1/3 of the tasks measuring the understanding and ability to interpret non-literal speech, proverbs and metaphors, the tasks measuring third-order ToM, as well as 1/3 of a dynamic emotion-recognition test (the TASIT—PART I: Emotion Evaluation Test) [56] were administered. At this point, it must be mentioned that the Natsopoulos’ battery includes tasks that have been developed for a specific cultural context based on the most representative tasks developed to measure adult ToM internationally, and the TASIT battery and especially the TASIT-PART 1 task is considered appropriate for international administration without the use of voice [55,56].

#### 2.7.2. Νon-Literal Speech: Humor, Irony, Sarcasm, and Faux Pas Understanding Stories

Participants were presented with four short vignettes designed to display ironic, humorous, sarcastic, and faux pas exchanges. Participants answered questions assessing their understanding of the non-literal meaning of each vignette. As regards Humor, Irony and Sarcasm comprehension, each story was scored with a maximum of 3 points: 1 point for the “Truth Question” (e.g., Is what [name] said true?; correct answer “No” = 1 point) (non-literal speech 1), and up to 2 points for the “Intention Question” (e.g., Why do you think [name] said that?) (non-literal speech 2). Scoring for the intention question included 0 points for no recognition of the non-literal meaning of the story, 1 point for identifying the non-literal meaning without correctly labeling it, and 2 points for both correct identification and labeling of the non-literal meaning. To ensure cognitive comprehension of the story’s content, control questions were also included (but were not scored). An example vignette is the following:

“Mary’s mother makes delicious homemade sweets. Mary asked her to prepare a special dessert for a friend who was coming to visit. When her mother showed her the sweet, Mary was so engrossed in playing a video game that she did not even turn to see or thank her. Her mother said to Mary, ‘Your kindness really moves me’”.

As regards faux pas comprehension, six questions are asked to evaluate their understanding of the key details. The participant is asked to identify whether they recognized the inappropriateness of the statement, determine the speaker’s intention, explain why such a statement should not have been made (e.g., rudeness, insult, ruining a surprise), and describe how the recipient of the comment felt (e.g., awkward, unpleasant). Correct responses are scored 1, and the maximum possible score is 6. However, in this analysis, only the first two questions identifying the inappropriate comment (non-literal speech 1) and understanding the speaker’s intention (non-literal speech 2) were included in the calculation of the score. Therefore, the total score could range from 0 to 2, reflecting the participant’s ability to recognize the inappropriateness of the comment and understand the speaker’s intent.

#### 2.7.3. Proverbs in Context

Four proverbs are presented with four interpretation options. One gives a literal meaning (scored 0), another provides an irrelevant interpretation (scored 0), while the other two offer correct interpretations. Only one is more accurate and complete, receiving a score of 2, while the other is scored 1. The maximum score on this test is 8. The participant must choose the most accurate interpretation of the proverb. Here is an example:

“Do good and throw it into the sea”.

When someone does good, they should throw it like a stone into the sea.When someone does a good deed, they should forget about it.Many people don’t care about the poor.Good deeds should not be advertised, and no reward should be expected.

#### 2.7.4. Verbal Metaphors in Context

Sentences containing metaphorical verbs are presented (four per test). Below each, three alternative sentences are provided, each rephrasing the original. The participant must choose the sentence that best captures the meaning of the original sentence. Only the correct answer is scored with 1 point. The maximum score is 4.

Example: “The famous card player stripped the casino’s bank”.

The famous card player won all the casino’s bank money.The famous card player undressed the casino’s bank.The famous card player impressed all the other players at the casino.

#### 2.7.5. Nominal Metaphors in Context

Sentences (4 in each test) containing metaphors in their nouns are provided. Below each sentence, three alternative interpretations are given, rephrasing the original sentence. The participant must choose the option that best conveys the meaning of the original sentence. Only the correct answer receives 1 point. The maximum score is 4. 

E.g., “Long-lasting friendship is wine”. The sentence means:Long-time friends often drink wine.Long-lasting friendship has a beneficial effect on people’s lives.Wine usually accompanies food.

#### 2.7.6. Third-Order ToM Stories

Third-order ToM was assessed using a “double bluff” story. This task evaluates advanced ToM reasoning, requiring participants to understand the thoughts of one character (Person X) about the thoughts and beliefs of another character (Person Y) regarding Person X’s intentions. Participants first answered control questions to confirm cognitive comprehension of the story. If they responded correctly, they were then asked whether the statement made by Person X in the story was true. If they answered affirmatively, two additional questions followed. One question required participants to infer the protagonist’s intention to deceive, achieved through the “bluff,” while the other required identifying the misunderstanding of this technique by the other characters in the story. Both the comprehension and justification questions, when answered correctly, were scored 1 point each. Control questions were not scored. The maximum possible score for the story was 3 points.

### 2.8. Visual Dynamic Emotion-Recognition Test

#### The TASIT—PART I: Emotion Evaluation Test (EET–PART 1–FORM A)

TASIT [56,57] was developed to assess an individual’s ability to recognize six fundamental emotions: happiness, pleasant surprise, sadness, anger, anxiety, and disgust, and to distinguish these from neutral expressions as they are dynamically portrayed by professional actors. The test consisted of 9 different short video clips (lasting 15–60 s) showing people interacting in everyday scenarios. After watching each clip without voice, participants are asked to select the emotion expressed by the actor from a list of six emotional categories and a neutral option, displayed randomly on one of five Response Cards. The content of the scripts is neutral and does not suggest any particular emotion. Given that the original test was developed in English, the current study administered it with the sound turned off to focus on the participant’s ability to interpret dynamic visual cues from expressions and gestures [57]. For the present study, scoring involves calculating a total score based on correctly identifying each of the six emotions or the neutral expression. The maximum score is 9.

### 2.9. Cognitive Planning

Delis–Kaplan Executive Function System (DKEFS) is a battery that provides an assessment of higher-order cognitive functions supported by the frontal lobe (executive functions) [58]. For the broader study, we chose to assess the main cognitive control abilities, namely inhibitory control, set-shifting, cognitive flexibility, and planning, with the use of specific tests of the DKEFS. In this study, only cognitive planning is a variable of interest that was measured by the DKEFS Tower Test Standard Form—DKEFS–TT, SF, which is a tool designed to assess complex executive functions such as spatial planning, rule learning, inhibition, and the ability to establish and maintain cognitive sets. The three Tower Test scores computed were the following: (i) Total Number of Problems Solved: Reflecting problem-solving and planning abilities; (ii) Total Number of Rules’ Violations: Highlighting deficits mainly in inhibitory control; and (iii) Total Achievement Score: A composite metric of overall task performance, integrating planning, adherence to rules, and problem-solving.

### 2.10. Sleep Duration

For the broader study, we administered a series of self-report questionnaires to the participants to assess various sleep-related parameters, including insomnia severity, sleep apnea risk, and sleep quality. These tools have been described in detail in a previous article [36], which found that no differences in these variables exist between the groups of the study cross-sectionally.

#### Actigraphy

Actigraphy is a method used to estimate sleep–wake patterns by analyzing movement data [59]. It is commonly utilized in research due to its potential to monitor activity in real-world settings. All participants in the study wore an actigraphy device for seven days. This wrist-worn device, resembling a watch, records movement data that, when processed, provides insights into the sleep–wake cycle. We utilized the Philips Respironics Actiwatch Spectrum Pro (version 5.57.0006) for this purpose. The device tracks motion and light exposure, offering valuable information on activity levels, sleep patterns, and sleep quality. Actigraphy data are useful in sleep medicine for assessing sleep disorders, circadian rhythm disruptions, and daily activity [60]. Sleep parameters like sleep onset, duration, awakenings, and efficiency are automatically calculated and displayed through a basic reader, which allows data visualization on a computer or tablet [61]. In the context of this study, and based on findings [36], sleep duration as total sleep time (TST) emerged as the only critical variable. TST was the only parameter to demonstrate statistically significant differences across the study groups, with shorter durations observed in individuals with naMCI and aMCI, compared to healthy controls and between each other.

### 2.11. Statistical Analysis

Statistical analysis was performed using IBM SPSS Statistics, Version 27 [62]. To test whether the three groups differed in ToM performances, the following analyses were conducted: (a) multivariate analysis of variance (MANOVA) and (b) one-way ANOVA. To control for multiple tests, Bonferroni correction was applied. A *p*-value < 0.007 was considered indicative of statistical significance for the ToM performances, i.e., *p* = 0.05/7 (dependent variables) = 0.007. Post hoc comparisons were conducted using the Scheffe test. After the computation of correlations of the variables of interest, mediation analysis within the context of Structural Equation Modeling (path models with mediators) was sequentially conducted using JASP 16 [63] to examine whether TST (the predictor) directly or/and indirectly—via the three DKEFS–TTscores (the mediators)—affects performance in ToM tests (outcome variables). Given that the N of the variables included in the mediation model was 11, we considered that the appropriate N of participants, in order to run the analysis, should be at least 110.

## 3. Results

### 3.1. ToM Performance

MANOVA was conducted to investigate the differences in ToM performance between the three groups on the ΤοΜ tests. As dependent variables, the seven ToM scores were used. The diagnostic group, with three levels (HC, aMCI and naMCI), was identified as the independent variable.

The results indicate that the diagnostic group had a significant effect, F (14.342) = 16.041, *p* < 0.001, η^2^ = 0.39, on the dependent variables. In particular, the diagnostic group effect was significant for (i) Non-literal Speech 1, F = (2,17) = 47.20, *p* < 0.001, η^2^ = 0.34, (ii) Non-literal speech 2, F = (2,17) = 137.18, *p* < 0.001, η^2^ = 0.60, (iii) Τhird-order ToM stories, F = (2,17) = 7257, *p* < 0.001, η^2^ = 0.45, (iv) Proverbs in Context, F = (2,17) = 31.35, *p* < 0.001, η^2^ = 0.26, (v) Nominal Metaphors in Context, F = (2,17) = 20.42, *p* < 0.001, η^2^ = 0.18, and (vi) TASIT—PART 1, F = (2,17) = 22.10, *p* < 0.001, η^2^ = 0.20. However, no statistically significant differences were observed in the performance of the three groups on the Verbal Metaphors in Context test. For this reason, this test is excluded from Figure 1.

Scheffe’s post hoc comparisons revealed significant differences in ΤοΜ tests among the three groups. For non-literal Speech 1, which measures simple recognition of non-literal speech, HC demonstrated significantly higher performance than the naMCI group, I-J = 1.34, *p* = 0.001. Additionally, the aMCI group also outperformed the naMCI group in this task, I-J = 0.85, *p* = 0.001.

In the more complex non-literal speech 2 conditions, which assess the intention of non-literal speech, the HC group strongly outperformed both the aMCI, I-J = 2.48, *p* = 0.001, and naMCI groups, I-J = 6.45, *p* = 0.001, indicating a marked superiority in understanding subtle social cues. The aMCI group also showed significantly higher performance compared to the naMCI group in this task, I-J = 3.97, *p* = 0.001, suggesting that the aMCI group retains a greater ability to grasp non-literal speech than the naMCI group, particularly in more advanced social reasoning tasks.

Significant differences were found in the performance on the Τhird-order ToM stories task across all three groups. The HC group demonstrated significantly higher scores compared to both the aMCI group, I-J = 1.03, *p* = 0.001, and the naMCI group, I-J = 1.71, *p* = 0.001, indicating a stronger ability to understand others’ mental states and perspectives. Furthermore, significant differences were observed between the two subtypes of MCI, with the aMCI group performing higher than the naMCI group, I-J = 0.68, *p* = 0.001. This suggests that individuals with amnestic MCI retain a greater capacity for complex social cognition compared to those with non-amnestic MCI.

The Proverbs in Context task revealed significant differences in scores as well. In particular, the HC group achieved higher scores compared to both the aMCI group, I-J = 1.04, *p* = 0.001, and the naMCI group. I-J = 1.92, *p* = 0.001. This finding indicates that healthy older adults are more adept at interpreting proverbs within contextual frameworks, demonstrating a higher understanding of nuanced language and cultural references. Moreover, significant differences were observed between the two subtypes of MCI. Specifically, the aMCI group outperformed the naMCI group, I-J = 0.88, *p* = 0.001, suggesting that individuals with aMCI possess a greater ability to comprehend proverbs in context compared to those with naMCI.

The Nominal Metaphors in Context assessment revealed variations in the performance among the groups, with HC demonstrating superior abilities compared to both the aMCI group, I-J = 0.78, *p* = 0.001, and the naMCI group I-J = 0.98, *p* = 0.001. These findings suggest that healthy older individuals possess a heightened capacity to comprehend nominal metaphors within contextual settings, reflecting their advanced understanding. In contrast, the performance between the aMCI and naMCI groups did not show statistically significant differences.

Regarding emotion recognition, significant differences were identified among the groups. The HC exhibited superior performance compared solely to the naMCI group, with a difference of I-J = 0.23, *p* = 0.001. This finding indicates that healthy older adults are more adept at recognizing emotions compared to their naMCI counterparts. Furthermore, individuals with aMCI outperformed those with naMCI, as indicated by a difference of I-J = 0.14, *p* = 0.001. This suggests that individuals with aMCI maintain a relatively higher ability to comprehend emotions compared to those with naMCI.

Statistically significant differences were not observed in the performance of the three groups in the Verbal Metaphors in Context test.

### 3.2. Correlations Between All Variables of Interest

Pearson correlation coefficients were used to analyze the relationships between total sleep time, the three variables of cognitive planning, and each of the ToM variables: Non-literal Speech 1, Non-literal Speech 2, Proverbs in Context, Verbal Metaphors in Context, Nominal Metaphors in Context, third-order ToM stories, and TASIT-Part 1. Significant correlations were observed between total sleep time and performance on the third-order ToM stories (r = 0.244, *p* < 0.01), Non-literal Speech 2 (r = 0.188, *p* < 0.05), and Proverbs in Context (r = 0.192, *p* < 0.05) for the total sample. Additionally, significant correlations were found between total sleep time and performance on DKEFS–TT, SF, namely: (i) the total number of problems given (r = 0.169, *p* = 0.023), (ii) the total number of violations (r = −0.280, *p* < 0.001), and (iii) the total achievement score (r = 0.158, *p* = 0.034). Cognitive planning variables were significantly correlated with almost all the ToM variables (*p* < 0.05). However, relatively new suggestions about mediation analysis do not consider as absolute perquisition the direct correlation between the predictor and the outcome variable to proceed with mediation analysis [64]. Hence, we decided to include all ToM variables as outcome variables in the mediation model.

### 3.3. Mediation Analysis

To examine whether sleep duration, as well as cognitive planning levels, could also play a role in decreasing ToM abilities, a mediation model was structured for the total sample. TST was defined as the predictor variable, ToM performances were defined as the outcome variables, and DKEFS–TT scores were the mediators.

None of the direct effects of TST on the dependent variables was statistically significant. However, regarding the indirect effects of TST on the various dependent variables, through the mediator variables of DKEFS–TT, several statistically significant relationships were found (see Table 1).

## 4. Discussion

### 4.1. The Role of Amnestic and Non-Amnestic Mild Cognitive Impairment in Theory-of-Mind Performance

According to the first hypothesis (H1a), individuals with MCI (both aMCI and naMCI) were expected to present poorer ToM performances compared to healthy controls. Additionally, we anticipated differentiation between the subtypes of MCI in their performance, at least on specific ToM tasks, with the group of naMCI displaying the lower performance (H1b). The results confirm Hypothesis 1a, as significant differences were observed in the performance of individuals with MCI—both amnestic and non-amnestic types—compared to healthy controls, confirming other relevant studies [65,66]. In nearly all ToM tasks, HC consistently outperformed those with MCI. Both aMCI and naMCI groups exhibited lower performance, particularly in tasks involving non-literal speech and third-order ToM. In general, these findings align with existing literature, which highlights the negative impact of MCI on ToM performance [8,67]. MCI affects ToM due to difficulties arising from the dysfunction of neural circuits involved in processing social information, reducing the accuracy of understanding others’ mental states [14]. Among other effects, it leads to degeneration of the medial temporal lobe, including damage to the hippocampus and related regions, which impairs the ability to retrieve past information necessary for connecting with the intentions or emotions of others [68]. Additionally, it causes dysfunction in the prefrontal cortex, affecting introspection and mental simulation processes required for understanding social situations [68].

Our findings also confirm Hypothesis 1b, as individuals with aMCI performed higher than those with naMCI on most ToM tests. Understanding irony, humor, sarcasm, and faux pas forms of non-literal speech requires distinct processing steps of ToM. Classical models have assumed that the comprehension of non-literal speech involves at least two processing steps [55], i.e., computing the literal meaning first, rejecting it as contextually inappropriate (non-literal speech 1) and finally, reinterpreting the utterance and arriving at the intended meaning by assessing the intention (non-literal speech 2). Difficulties in simple recognition (non-literal Speech 1) among MCI participants reflect deficits in the ability to move beyond the literal meaning of statements and process contextual information [69]. These challenges were more pronounced in naMCI participants, who exhibited greater impairments compared to aMCI participants, probably due to more severe deficits in the underlying brain networks of complex ToM [14,68]. Similarly, difficulties in comprehending the intention of non-literal speech (non-literal Speech 2) may highlight the reduced capacity for metacognitive reasoning and mental state attribution, particularly in very complex social contexts. Non-literal speech comprehension relies on an integrated ToM capacity that encompasses both literal rejection and intention inference [70].

The findings of the present study also highlight significant differences in proverb interpretation among healthy controls, aMCI, and naMCI participants, underscoring the impact of a potential abstract reasoning and semantic processing decline [71]. Healthy controls consistently outperformed both MCI subgroups. The superior performance of healthy controls in proverb tasks can be attributed to the intact functioning of brain regions involved in abstract thinking, such as the dorsolateral prefrontal cortex and posterior temporal regions, which are responsible for semantic integration [72]. While aMCI participants exhibited higher performance than their naMCI counterparts, suggesting that aMCI may involve relatively preserved functioning of these regions, particularly in the early stages of the condition [66]. Proverb interpretation tasks, which require selecting the most accurate and contextually appropriate interpretation while rejecting literal or irrelevant meanings, place strong demands on semantic integration [73]. The relative advantage of aMCI participants may reflect a partial preservation of these abilities, whereas naMCI participants, who typically show more severe disruptions in frontal and temporal lobe circuits [8], may reflect more pronounced deficits in semantic control, struggle with the cognitive demands of these tasks [8,30].

Significant differences were revealed in the ability to interpret nominal metaphors as well, highlighting the challenges posed by metaphor comprehension in the context of cognitive decline [74]. Metaphor interpretation requires a robust interaction between the anterior temporal lobes, which are critical for semantic processing, and the prefrontal cortex, which is involved in the executive control of these processes [75]. Impairments in functional connectivity within the prefrontal cortex and DMN are hallmark features of MCI and profoundly affect the ability to understand intentions [76,77,78]. Healthy controls consistently outperformed MCI participants, potentially underscoring the role of intact semantic processing in understanding abstract linguistic constructs [74]. However, in this study, no significant differences were observed between aMCI and naMCI subgroups, suggesting that both subtypes experience comparable difficulties in processing nominal metaphors. Also, it may reflect a shared vulnerability in these brain regions across both subtypes. This aligns with findings that metaphor interpretation requires a preserved ability to integrate semantic and contextual cues [70]. Unlike other tasks where various types of deficits may disproportionately affect naMCI individuals, the uniform performance in this task suggests a shared vulnerability in semantic abstraction and associative reasoning among MCI subtypes. These results reinforce the diagnostic value of metaphor interpretation tasks in identifying early deficits in abstract language processing while highlighting the need for further exploration into the cognitive mechanisms underlying metaphor comprehension in different MCI profiles [79].

Emotion recognition using TASIT-Part I is a basic-level tool that assesses the ability to perceive fundamental emotions from dynamic visual stimuli. However, it cannot provide a comprehensive differentiation between MCI subtypes and healthy individuals, as it does not incorporate more complex aspects of social cognition and emotional processing. Therefore, more specialized tools are needed to fully understand the differences in emotion recognition.

On the contrary, the findings primarily emphasize the impact of MCI subtypes on advanced cognitive ToM reasoning, as demonstrated by performance on third-order ToM stories tasks. Specifically, healthy controls consistently outperformed both aMCI and naMCI participants, while aMCI participants demonstrated higher performance than naMCI participants. These results align with the literature, which highlights the critical role of social–cognitive capacities in complex ToM tasks such as understanding “double bluff” scenarios [8]. Third-order ToM stories tasks, which require individuals to interpret recursive mental states (e.g., Person X’s belief about Person Y’s interpretation of X’s intentions), place significant demands on working memory. The superior performance of aMCI participants could suggest partial preservation of working memory capacity compared to naMCI participants [30,66]. Τhe observed differences in this task performance across the groups can also be understood in the context of neurobiological changes associated with MCI. The hippocampal and prefrontal regions, often affected in MCI (especially in the naMCI subgroup), are critical for such higher-order cognitive abilities [80]. The current findings underscore the value of using advanced-level cognitive ToM tasks, such as those assessing Third-order reasoning, to detect nuanced differences in social cognition across MCI subtypes, as they provide a sensitive measure of complex cognition impairments [8].

In any case, the findings underscore that the mechanisms underlying ToM deficits in MCI are not uniform, varying according to the MCI subtype. This highlights the need for tailored interventions that address the specific social cognition challenges faced by individuals with aMCI and naMCI. In line with these observations, we propose including non-literal speech tasks (such as understanding humor, irony, sarcasm, and faux pas), third-order ToM stories, and proverb comprehension in context as part of a diagnostic battery for MCI and its subtypes. These tools assess the ability to understand complex social messages, recognize others’ intentions, and interpret symbolic meanings in social contexts, offering a comprehensive approach to differentiate MCI subtypes and facilitate early diagnosis and monitoring of the disease.

### 4.2. The Impact of the Associations Among Sleep Duration, Cognitive Planning and ToM Abilities

The findings of the present study provide some critical insights into the complex relationship between sleep duration, cognitive planning, and ToM performance. While reduced sleep duration was hypothesized to be directly associated with poorer ToM performance (H2a), the results revealed no statistically significant direct effects of TST on ToM tasks. However, the mediation analysis demonstrated significant indirect associations of TST and specific ToM tasks, mediated primarily by rules’ violations, a measure of cognitive planning that reflects difficulties in self-regulation and decision-making. These difficulties in cognitive planning, as indicated by rules’ violations, may hinder the ability to effectively interpret and respond to social cues, thus influencing performance on ToM tasks (H2b).

The absence of direct associations between sleep duration and ToM performances in the mediation model underscores the complex and multifactorial nature of social cognition [81]. ToM depends on an intricate network of neural and cognitive mechanisms, which may mitigate the potential immediate impact of sleep reductions [82]. For example, while sleep deprivation is known to impair cognitive processes like memory consolidation and executive function [83], certain ToM-relevant neural circuits, such as those in the medial prefrontal cortex and temporoparietal junction, may retain sufficient functionality to support performance on ToM tasks even in the context of sleep reduction [84]. This neural compensation could explain why ToM abilities may appear resistant to direct impacts from sleep loss.

However, mediation analysis revealed that mainly rules’ violations, a specific indicator of cognitive planning, emerged as a constant mediator in the relationships between sleep duration and all ToM aspects. Indeed, reduced sleep duration has been shown to compromise the ability to adhere to task rules, inhibit impulsive responses, and adapt to complex scenarios [85]. Rules’ violations affect ToM performance in several ways [84]. Rule violations indicate that cognitive control and self-regulation abilities are impaired. This means individuals struggle to follow the rules, inhibit impulsive reactions, and adapt to social or complex situations, which negatively impacts their ability to understand others’ mental states and perform ToM tasks.

As regards the non-literal speech tasks requiring the rejection of literal interpretations (non-literal speech 1) and the inference of speaker intentions (non-literal speech 2) demands, rules’ violations may represent inhibitory control problems at the language level. Such types of impairment hinder the capacity to process contextual information and infer complex social cues [70].

As regards third-order ToM tasks, especially, they involve recursive mental state reasoning, placing substantial demands on the ability to follow hierarchical rules [86]. Proverb interpretation further illustrates the potential role of rule violations in ToM performance. As said, this task requires abstract reasoning and the application of contextual rules to derive non-literal meanings. Rules’ violations may represent a reduced ability to effectively integrate semantic and contextual cues [87]. Similarly, nominal metaphor comprehension sheds light on the potentially subtle effects of rule violations [88]. Again, challenges in abstract semantic processing reflect a shared vulnerability in associative reasoning. This shared vulnerability suggests that impairments in integrating abstract and contextual information may underlie broader deficits observed across non-literal language tasks [89].

Future research should explore the longitudinal dynamics of sleep, cognitive control, and ToM abilities across MCI subtypes, focusing on causal pathways and neural mechanisms. Such studies could guide personalized therapeutic approaches, addressing both cognitive and social challenges in MCI.

## 5. Conclusions

In conclusion, performance on specific but the most representative, complex aspects of ToM, such as non-literal speech comprehension (sarcasm, humor, irony and faux pas understanding) and third-order ToM stories understanding, provides clear differentiation between healthy aging and amnestic MCI and non-amnestic MCI pathologies and holds potential as diagnostic tools. Besides MCI pathologies, the most significant cognitive impairment linked to lower ToM performance is related to rule violations in cognitive planning, which reflect serious difficulties in understanding or managing social norms and behaviors. Notably, rules’ violations in planning are associated with decreasing sleep duration, suggesting that factors such as sleep may contribute to the decline in ToM abilities, in addition to the pathology of MCI.

These findings underline the importance of addressing ToM deficits, in MCI and especially in naMCI, not only for diagnostic purposes but also for informing caregivers and healthcare providers, enabling the development of targeted interventions to improve social interactions and quality of life for affected individuals.

## 6. Limitations

In our study, actigraphy was utilized to observe the sleep patterns of our sample. While it provided valuable insights due to its affordability and naturalistic data collection, its limitations, such as the inability to detect detailed sleep architecture (e.g., REM, NREM) and reliance on movement, were acknowledged. Despite these constraints, it proved to be a practical tool for our research objectives. Additionally, the lack of structural MRI and biomarkers, such as amyloid-β and PET scans, reduces the ability to link behavioral findings to neurobiological changes and prevents a clear connection between the variables of interest and biological alterations in individuals with MCI. Furthermore, the cross-sectional design of the study restricts the ability to monitor changes in the variables of interest over time, hindering the observation of developmental trajectories. It is worth mentioning that a longitudinal study has been completed, which will offer additional insights into these trajectories.

## 7. Future Implications

Longitudinal studies are essential to clarify which aspects of ToM are most affected in MCI and differentiate its subgroups to fully understand the impact on social–cognitive function during aging. These studies will help identify the specific areas of ToM that are impaired by MCI and assist in classifying subgroups, allowing for a better understanding of differences in social interaction and social perception among individuals with mild cognitive impairment. This research highlights the importance of conducting longitudinal studies that assess sleep duration over extended periods, as well as valid interventions aimed at improving sleep, to clarify the complex relationships between habitual sleep patterns and social cognition in both healthy and pathological aging.

## Figures and Tables

**Figure 1 brainsci-15-00057-f001:**
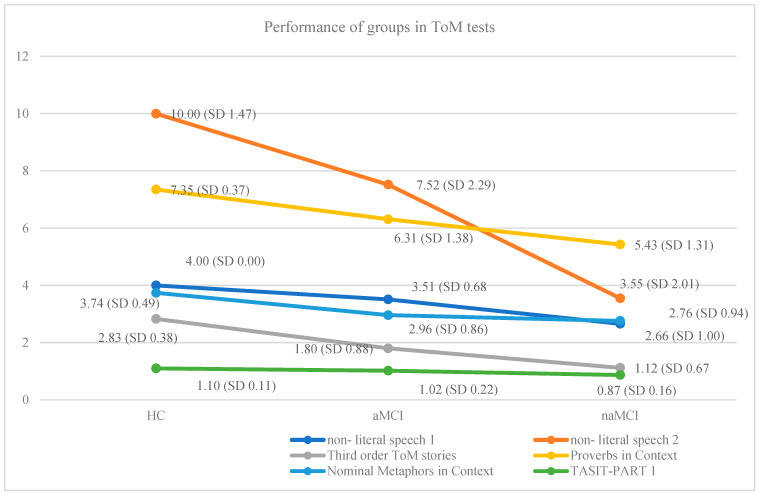
The effects of diagnostic group on six ToM Tests.

**Table 1 brainsci-15-00057-t001:** Direct and Indirect Effects of Sleep Duration (TST) on the ToM tasks, including the Mediating Role of three Cognitive Planning Scores.

Pathway	Estimate	Std. Error	z-Value	*p*	95% CI Lower	95% CIUpper
**Direct effects**						
TST → TASIT-PART 1	0.113	0.180	0.630	0.529	−0.239	0.466
TST → non-literal speech 1	−0.399	0.747	−0.534	0.593	−1.862	1.065
TST → non-literal speech 2	1.312	2.420	0.542	0.588	−3.431	6.054
TST → third-order ToM stories	1.270	0.759	1.674	0.094	−0.217	2.758
TST → Nominal Metaphors	−0.840	0.804	−1.045	0.296	−2.416	0.735
TST → Proverbs in Context	1.759	1.252	1.405	0.160	−0.695	4.213
TST → Verbal Metaphors	0.092	0.662	0.139	0.889	−0.206	1.390
**Indirect effects**						
TST → DKEFS–TT Total achievement score → TASIT-PART 1	0.007	0.046	0.153	0.878	−0.084	0.098
TST → DKEFS–TT Total number of problems → TASIT-PART 1	0.112	0.069	1.628	0.104	−0.023	0.248
**TST** → **DKEFS–TT Total number of rules’ violations** → **TASIT-PART1**	**0.113**	**0.056**	**2.026**	**0.043**	**0.004**	**0.223**
TST → DKEFS–TT Total achievement score → Non-literal speech 1	−0.457	0.287	−1.593	0.111	−1.020	0.105
**TST** → **DKEFS–TT Total number of problems** → **Non-literal speech 1**	**0.748**	**0.383**	**1.951**	**0.051**	**−0.003**	**1.499**
**TST** → **DKEFS–TT Total number of rules’ violations** → **Non-literal speech 1**	**1.058**	**0.380**	**2.786**	**0.005**	**0.314**	**1.802**
TST → DKEFS–TT Total achievement score → Non-literal speech 2	−1.012	0.782	−1.295	0.195	−2.544	0.520
**TST** → **DKEFS–TT Total number of problems** → **Non-literal speech 2**	**2.719**	**1.353**	**2.010**	**0.044**	**0.068**	**5.371**
**TST** → **DKEFS–TT Total number of rules’ violations** → **Non-literal speech 2**	**4.530**	**1.548**	**2.927**	**0.003**	**1.496**	**7.563**
TST → DKEFS–TT Total achievement score → Τhird-order ToM stories	0.179	0.212	0.842	0.400	−0.238	0.595
TST → DKEFS–TT Total number of problems → Τhird-order ToM stories	0.369	0.261	1.414	0.157	−0.142	0.880
**TST** → **DKEFS–TT Total number of rules’ violations** → **Τhird-order ToM**	**1.114**	**0.397**	**2.807**	**0.005**	**0.336**	**1.892**
TST → DKEFS–TT Total achievement score → Nominal Metaphors	0.159	0.220	0.723	0.470	−0.272	0.590
TST → DKEFS–TT Total number of problems → Nominal Metaphors	0.315	0.257	1.223	0.221	−0.190	0.819
**TST** → **DKEFS–TT Total number of rules’ violations** → **Nominal Metaphors**	**0.592**	**0.268**	**2.207**	**0.027**	**0.066**	**1.118**
TST → DKEFS–TT Total achievement score → Proverbs in Context	−0.137	0.329	−0.416	0.678	−0.781	0.507
TST → DKEFS–TT Total number of problems → Proverbs in Context	0.798	0.486	1.644	0.100	−0.153	1.750
**TST** → **DKEFS–TT Total number of rules’ violations** → **Proverbs in Context**	**0.970**	**0.429**	**2.264**	**0.024**	**0.130**	**1.810**
TST → DKEFS–TT Total achievement score → Verbal Metaphors	0.176	0.189	0.930	0.352	−0.195	0.547
TST → DKEFS–TT Total number of problems → Verbal Metaphors	0.285	0.218	1.306	0.192	−0.143	0.713
TST → DKEFS–TT Total number of rules’ violations → Verbal Metaphors	0.006	0.158	0.037	0.971	−0.304	0.315

Bold values indicate statistically significant results (*p* < 0.05).

## Data Availability

The data presented in this study are available on request from the corresponding author, as they are part of my doctoral dissertation, which has not yet been fully published.

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
