# Peer review of "Examining Specific Theory-of-Mind Aspects in Amnestic and Non-Amnestic Mild Cognitive Impairment: Their Relationships with Sleep Duration and Cognitive Planning"

_brainsci, 2025, doi:10.3390/brainsci15010057_

Round 1
Reviewer 1 Report
Comments and Suggestions for Authors
The aim of the manuscript entitled "Examining specific Theory of Mind aspects in Amnestic and Non-Amnestic Mild Cognitive Impairment: Their Relationships with Sleep Duration and Cognitive Planning" was to examine potential associations between sleep duration, various ToM abilities, and cognitive planning in Amnestic (aMCI) and Non-Amnestic (naMCI) Mild Cognitive Impairment subtypes. The results have identified significant differences in various ToM tasks’ performance between the groups, particularly in non-literal speech tasks and third-order ToM stories.
The manuscript is well written and easy to follow. The authors sought to enhance the existing knowledge about this research topic. However, minor changes are needed to improve the quality of the manuscript. The authors need to address these issues.
ABSTRACT
1) It is recommended that the potential social implications of the study be demonstrated.
INTRODUCTION
2) The link between ToM deficits and their practical implications (e.g., daily life challenges) could be improved by providing more contextual information. Furthermore, the theoretical framework is deficient in terms of detail. In order to enhance the quality and accuracy of the manuscript, it is recommended that other studies and systematic reviews on this topic be included. Please cite these articles: https://doi.org/10.3390/ijerph19053097; 10.1002/gps.4398)
3) In my opinion, the hypotheses are well-articulated, offering a focused aim for the study.
METHODS
4) The lack of biomarkers (e.g., amyloid-β) and neuroimaging data reduces the ability to link behavioral findings to neurobiological changes. I add this information in the limitation section.
5) L. 317: It is hypothesized that an error has occurred during the editing process.
RESULTS
6) The lack of direct effects of sleep on ToM performance is an area that has been underexplored, leaving unanswered questions about alternative pathways. The authors are invited to provide an explanation for this choice.
7) It is evident that the caption of Table 1 is deficient in certain details (L:522). It is recommended that the caption be rendered more informative.
8) It is imperative to elucidate the implications of the assertion that direct effects are non-significant, yet numerous significant indirect effects are in evidence. What theoretical frameworks underpin your use of mediation? In my opinion the interpretation of the mediation models is erroneous in several ways. According to the theoretical assumptions of statistical mediation, as articulated by Baron and Kenny (1986) and Gallucci and Leone (2017), the interpretation of the mediation models is erroneous. In the absence of a significant effect between the predictor and the dependent variable, the mediation assumption is considered violated.
DISCUSSION
9) There is minimal discussion of potential confounders (e.g. gender or educational disparities).
10) Conclusions could expand on the potential implications for caregivers or healthcare providers.
Author Response
Dear Reviewer,
We deeply appreciate the time you invested to review such thoroughly our manuscript. Your comments and suggestions were invaluable to substantially improve our manuscript. We tried to address all your comments. We trust that the changes we have made align with your recommendations as closely as possible. Below you can find a brief response to each of your points. All changes are highlighted in orange ink.
(Abstract)
Point 1: It is recommended that the potential social implications of the study be demonstrated.
Response 1: Thank you for your comment; we have incorporated the corresponding addition into the abstract, lines 42-45.
Introduction
Point 2: The link between ToM deficits and their practical implications (e.g., daily life challenges) could be improved by providing more contextual information. Furthermore, the theoretical framework is deficient in terms of detail. In order to enhance the quality and accuracy of the manuscript, it is recommended that other studies and systematic reviews on this topic be included. Please cite these articles: https://doi.org/10.3390/ijerph19053097; 10.1002/gps.4398)
Response 2: Thank you very much for your insightful comment. We have incorporated additional contextual information regarding the practical implications of ToM deficits and expanded the theoretical framework in the introduction. Furthermore, we have included the references you suggested to enhance the quality and accuracy of the manuscript, lines 60-67.
Point 3: In my opinion, the hypotheses are well-articulated, offering a focused aim for the study.
Response 3: Thank you for your comment.
Methods
Point 4: The lack of biomarkers (e.g., amyloid-β) and neuroimaging data reduces the ability to link behavioral findings to neurobiological changes. I add this information in the limitation section.
Response 4: We added this information in the limitations section, lines 741- 744.
Point 5: L. 317: It is hypothesized that an error has occurred during the editing process.
Response 5: Corrected
Results
Point 6: The lack of direct effects of sleep on ToM performance is an area that has been underexplored, leaving unanswered questions about alternative pathways. The authors are invited to provide an explanation for this choice.
Response 6: Thank you for your observation. The lack of direct effects is discussed in the subsection The Impact of the associations among sleep duration, cognitive planning and ToM abilities (pp. 16-17).
Point 7: It is evident that the caption of Table 1 is deficient in certain details (L:522). It is recommended that the caption be rendered more informative.
Response 7: The table caption has been rephrased to be more detailed and clearer, lines 557-558.
Point 8: It is imperative to elucidate the implications of the assertion that direct effects are non-significant, yet numerous significant indirect effects are in evidence. What theoretical frameworks underpin your use of mediation? In my opinion the interpretation of the mediation models is erroneous in several ways. According to the theoretical assumptions of statistical mediation, as articulated by Baron and Kenny (1986) and Gallucci and Leone (2017), the interpretation of the mediation models is erroneous. In the absence of a significant effect between the predictor and the dependent variable, the mediation assumption is considered violated.
Response 8: Thank you for raising important points regarding the interpretation of our mediation models and the theoretical framework guiding our analysis. To clarify, we actually did observe some significant effects in our initial correlations analyses (Pearson). These findings partially validate the relationships that are further explored in the mediation model. We will incorporate this evidence into the results section of the manuscript to provide transparency regarding the foundational analyses underpinning our mediation approach. However, we do not agree with the necessity of a direct relationship between predictor and outcome variable as an absolute prerequisite to proceed with mediation, since this point of view has been revised due to various reasons (please see the indicative reference we added), lines 543-545.
Our mediation analysis builds on these initial findings to explore potential pathways through which sleep duration (predictor variable) influences ToM performance (outcome variables), with particular attention to cognitive planning as a mediator. While the direct effects may not remain significant in all models, the robust indirect effects highlight potential mechanisms at play which are discussed in pp 16-17.
Discussion
Point 9: There is minimal discussion of potential confounders (e.g. gender or educational disparities).
Response 9: The groups are balanced in terms of gender and education, ensuring that these factors do not pose potential confounders in our study. More information is provided in the section regarding the participants, lines 229-246.
Point 10: Conclusions could expand on the potential implications for caregivers or healthcare providers.
Response 10: Thank you for your suggestion. We have incorporated some potential implications for caregivers and healthcare providers into the conclusions, lines 735-738.
Thank you once again for your constructive feedback.
Kind regards.

Reviewer 2 Report
Comments and Suggestions for Authors
Dear Authors,
This is a really intresting study that delves into investigating the interplay between TOM, sleep and cognitive planning in individuals with MCI. However, there are some issues raised in order for this study to be ready to be published.
According to clarity issues, the authors should deliver a more clear distinction between results, interpretation and their implications.
Methodologically, the authors should aknowledge more clearly the limitation of acitigraphy to detect sleep achitecture compared to polysomnography. In addition to this the authors should also discuss how the final sample size aligns with the power analysis. Last but not least the authors should discuss better why specific TOM tasks were selected for analysis.
According to Results, authors could discuss more about several findings like no significant direct effects of sleep on TOM tasks but highlighted indirect effects though cognitive planning. This could be benefited by a deeper analysis of potential nruobiological mechanisms. Further more, the lack of significant differences in verbal metaphors in context still requires further investigation. Is this a case of a methodological or conceptual limitation in task sensitivity?
Overall this is a high-quality study. After the revisions it should be ready to be published.
Author Response
Dear Reviewer,
We sincerely appreciate the time and effort you dedicated to reviewing our manuscript in such detail. Your insightful comments and suggestions have been invaluable in enhancing the quality of our work. We have made every effort to address all your points. We believe that the revisions we have implemented align as closely as possible with your recommendations. Below, we provide a brief response to each of your comments. All changes are highlighted in orangeink.
Point 1: According to clarity issues, the authors should deliver a more clear distinction between results, interpretation and their implications.
Response 1: Thank you for your feedback. We understand the importance of making a clear distinction between the results, interpretation, and implications in our manuscript. In response to your comment, we have revised the sections to better delineate these aspects. We have made changes in various sections of the manuscript to ensure that the results are clearly presented, the interpretation is distinct, and the implications are properly discussed. Please review these changes for clarity and coherence.
Point 2: Methodologically, the authors should acknowledge more clearly the limitation of actigraphy to detect sleep architecture compared to polysomnography. In addition to this the authors should also discuss how the final sample size aligns with the power analysis. Last but not least the authors should discuss better why specific TOM tasks were selected for analysis.
Response 2: We have enriched the limitation section by providing more details on the limitations of actigraphy in detecting sleep architecture compared to polysomnography, lines 740-744.
As mentioned in the Participants section, power analysis was conducted using G*Power, and it revealed that a minimum of 148 participants was required to achieve a power level of 0.80. Τhe final sample consisted of 179 participants. This sample size is consistent with the power analysis, ensuring that the study had sufficient power to detect meaningful effects, lines 230-233. We also followed the rule 1/10 for the ratio variable/participants’ values for the more complex analysis (the mediation model).
Finally, we have selected the most widely used, representative tasks in the literature for assessing ToM, as these tasks are commonly employed in studies within this field. In addition, we aimed to include a broad range of tasks to capture the different aspects of ToM, ensuring a comprehensive analysis that reflects the diversity of ToM performance, lines 326-330.
Point 3: According to Results, authors could discuss more about several findings like no significant direct effects of sleep on TOM tasks but highlighted indirect effects though cognitive planning. This could be benefited by a deeper analysis of potential neurobiological mechanisms. Furthermore, the lack of significant differences in verbal metaphors in context still requires further investigation. Is this a case of a methodological or conceptual limitation in task sensitivity?
Response 3: Thank you for the suggestion. The lack of direct effects and the role of indirect effects is discussed in the subsection, The Impact of the associations among sleep duration, cognitive planning and ToM abilities (pp. 16-17).
The lack of significant differences in verbal metaphors in context is not a case of a methodological or conceptual limitation in task sensitivity. It is likely that verbal metaphors in context are simply easier to process, and as a result, they may not be affected by the pathology of MCI yet.
Thank you once again for your constructive feedback.
Kind regards.
